# Comparative Investigation of Raw and Processed Radix Polygoni Multiflori on the Treatment of Vascular Dementia by Liquid Chromatograph−Mass Spectrometry Based Metabolomic Approach

**DOI:** 10.3390/metabo12121297

**Published:** 2022-12-19

**Authors:** Fengye Wu, Yunlin Li, Wenya Liu, Ran Xiao, Benxing Yao, Mingzhe Gao, Di Xu, Junsong Wang

**Affiliations:** Center of Molecular Metabolism, Nanjing University of Science and Technology, Nanjing 210094, China

**Keywords:** Chinese materia medica processing, Radix Polygoni Multiflori, vascular dementia, metabolomics, HPLC−MS/MS

## Abstract

Radix Polygoni Multiflori (PM) is a well−known nootropic used in traditional Chinese medicine (TCM). Considering the efficacy and application discrepancy between raw (RPM) and processed PM (PPM), the similarities and differences between them in the treatment of vascular dementia (VaD) is intriguing. In this study, a VaD rat model was constructed by 2−vessel occlusion (2−VO). During 28 days of treatment, plasma was collected on days 7, 14, 21, and 28 after the start of dosing and the metabolic profile was analyzed by HPLC−MS/MS−based metabolomics. The Morris Water Maze Test, hematoxylin–eosin and Nissl staining, and biochemical analysis were used to assess cognitive function, pathogenic alterations and oxidative stress, respectively. RPM and PPM effectivelyreducedthe 2VO−induced cognitive impairment and mitigated histological alterations in hippocampus tissue. The 2−VO model significantly elevated MDA level and decreased SOD activity and GSH level, indicating severe oxidative stress, which could also be attenuated by RPM and PPM treatment. RPM outperformed PPM in decreasing MDA levels while PPM outperformed RPM in increasing GSH levels. Differential metabolites were subjected to Metabolite Set Enrichment Analysis (MSEA) and genes corresponding to proteins having interactions with metabolites were further annotated with Gene Ontology (GO). Both RPM and PPM ameliorated VaD−relevant vitamin B6 metabolism, pentose phosphate pathways, and taurine and hypotaurine metabolism. In addition, the metabolism of cysteine and methionine was regulated only by RPM, and riboflavin metabolism was modulated only by PPM. The results suggested that raw and processed PM had comparable efficacy in the treatment of VaD but also with some mechanistic differenece.

## 1. Introduction

Vascular dementia (VaD) is a syndrome of acquired and persistent intellectual impairment resulting from brain dysfunction caused by various cerebrovascular diseases. As the most common type of dementia after Alzheimer’s disease (AD), VaD manifests as a decline in memory, calculation, orientation and judgment, accompanied by affective disorder, behavioral abnormality, cognitive dysfunction and neurological dysfunction [1]. The risk factors for VaD include aging, hypertension, hyperhomocysteinemia, hyperlipidemia, recurrent stroke, cardiac disease, smoking, sleep apnea and diabetes mellitus [2]. Currently, treatment strategies for vascular dementia (VaD) focus on the control of underlying vascular risk factors and associated symptoms such as memantine (NMDA−R antagonist), donepezil and galantamine (anticholinergic agents). In addition, herbal active ingredients such as stigmasterol and vincristine can also be used to treat VaD [3]. Considering multiple aspects implicated in the pathogenesis of VaD, such as oxidative stress, excitotoxicity and inflammation [4], multi−component multi−target herbal medicine offers more possibilities than single−action therapy for the prevention and treatment of VaD and therefore is gaining increasing attention.

Based on “Medical Insights”, a classical traditional Chinese medicine (TCM) works, the kidney governs intelligence and deficiency of the kidney leads to insufficient intelligence [5]. The onset of dementiais closely related to kidney deficiency, and modern clinical practice for treatment of VaD is also based on tonifying the kidney [6]. Radix Polygoni Multiflori (PM) is the root of *Polygonum Multiflorum* Thunb. [7] and wellknown for its tonic effect. Herb frequency analysis of 92 TCM prescriptions for the treatment of VaD revealed that PMis among the top two TCM herbs with the highest use frequency of 61 [8]. A major component of PM, 2,3,5,4′−tetrahydroxy stilbene 2−O−β−D−glucoside (TSG), exhibited definite treatment effects on dementia [9,10]. 

In the Chinese Pharmacopoeia Commission [11], the raw (RPM) and processed (PPM) products of PM have different applications. RPM is used to eliminate carbuncles and treat malaria. With antioxidant [12], antiviral [13], anti−inflammatory [14], anti−aging and anti−atherosclerosis [15], and neuroprotection activities [16], RPM has been used for the prevention of cardiovascular [17] and neurodegenerative diseases [18]. Previous studies in our laboratory showed that the ethanol extract of RPM could alleviate ischemia−reperfusion brain injuries [7]. RPM is usually processed by steaming or boiling with a black bean decoction to produce PPM, the most widely−used tonic herb in TCM, which protects the liver and kidneys, nourishes essence and blood, darkens beard and hair, and strengthens tendons and bones [11]. PPM also has antioxidant, anti−aging [19] and cognitive function improvement effects [20]. Scopolamine−induced memory loss and cognitive impairment were alleviated by PPM ethanol extract, which regulates cholinergic function, brain−derived neurotrophic factor (BDNF) and neuronal death [21]. Therefore, both RPM and PPM have certain therapeutic effects on neurodegenerative diseases and cognitive deficits. 

In addition, it is generally believed that the tonic effects of processed PM are better than raw PM. This study evaluated whether there is aresimilarities or differences between raw and processed PM in the treatment of VaD by a holistic metabolomics approach.

## 2. Results

### 2.1. RPM and PPM Improved Memory Performance of VaD Rats

To evaluate the effects of treatment on the growth of the SD rats, their weight was recorded daily, and plotted in Figure 1A. There was no significant difference between groups, indicating that VaD and drug administration had no obvious influence on the body weight of the rats.

We used MWM to assess the effect of RPM and PPM on memory impairment. The spatial probing test and the location navigation test were the two procedures that made up MWM training. In the place navigation test, rats completed 4 training trials per day for 5 consecutive days to locate the underwater platform, using visual cues installed around the room. On training days 4 and 6, all rat groups displayed much shorter escape latencies than they had on training day 1, indicating fast learning of the escape platform location (Figure 1B). The escape distance and latency of the rats in M group were significantly longer than those of the rats in S group in the first 3 days (*p* < 0.05), with the treatment group in between, indicative of poor performance of spatial working memory by modeling and amelioration of the treatment. In the space exploration experiment, the difference in mean swimming speed was not significant among groups (*p* > 0.05), indicating that modeling and drug administration did not cause motor deficiency in rats (Figure 1D). In the spatial probe test, rats in M group spent less time in the target quadrant and had significantly fewer platform traversals than S group, indicating that 2−VO impaired cognitive function in rats. Compared with the M group (Figure 1E,F), the rats in the RPM and PPM groups showed no obvious change in time spent in the target quadrant (*p* > 0.05), but the frequency of crossing the platform area significantly increased (*p* < 0.001). The results (Table 1 and Table 2) indicated that RPM and PPM could improve the memory performance of VaD rats.

### 2.2. RPM and PPM Reduced the Histopathological Damage of VaD

The hippocampus is the most crucial component of the brain for memory and learning. It is divided into the dentate gyrus (DG), and the cornu ammonis (CA) regions (CA1, CA2, and CA3). The CA1 region is incredibly susceptible to ischemia injury and has a high link with cognitive function [22]. Compared with the S group rats, the hippocampal pyramidal cell layer of M group was significant thinned (Figure 2); the cells were loosely arranged; and some neuronal somas were deformed with enlarged gaps, indicating that the hippocampal structure was damaged and cell death occurred in the group. The hippocampal structure of the RPM and PPM groups was complete and neatly arranged, but some individual cells appeared deformed and vacuolated. To assess the loss of neurons in each group, histopathological abnormalities were detected by Nissl staining (Figure 2B) and the number of surviving neurons in hippocampal CA1 was measured by using Image J. The 2VO rats had fewer surviving neurons in CA1 compared to sham rats (*p* < 0.01). However, rats in the RPM and PPM groups had more surviving neurons than the 2VO rats (*p* < 0.05). The results indicated that both RPM and PPM gave a certain protection against the pathological changes of the hippocampus in VaD rats.

### 2.3. RPM and PPM Exhibits Anti−Oxidant Effects

It has been demonstrated that oxidative stress plays an important role in the etiology of VaD, and therefore, SOD activity and levels of MDA and GSH in serumwere measured (Figure 3). The SOD activity in the S group was significantly higher than the M group (*p* < 0.01), which was not changed after the treatment. As compared with the S group, the level of MDA was significantly increased (*p* < 0.001), which could be markedly decreased by treatment, especially by RPM (*p* < 0.001). GSH was significantly decreased in the M group (*p* < 0.001) and enhanced significantly by PPM (*p* < 0.01).

### 2.4. RPM and PPM Ameliorated Disordered Metabolism of VaD

In the Principal Component Analysis (PCA) score plot (Figure 4), S group and M group rats were well separated at T1, T2 and T4, indicating a significant modeling effect on the metabolism of rats at the above time points. The PCA score plot showed a clear time course effect, the administration groups at week 1 were close to M group and gradually approached to S group over time, indicating that RPM and PPM had some therapeutic effects on VaD in the long run.

### 2.5. Differential Metabolites in Treatment Group

The metabolites were identified by database matching of secondary mass spectrometry or putatively based on comparison of the accurate *m/z* values with monoisotopic mass of metabolites in the HMDB, KEGG and CHEBI databases (within a mass error of 20 ppm). Significantly differential metabolites between the treatment/sham group and the model group were screened based on the VIP (variable importance in the projection) values obtained from the PLS analysis. We reasoned that a high VIP value of a metabolite between several consecutive time points could be more promising as biomarker. Considering the proximity of the sham and model groups at T3 (21st day) in the PCA score plot (Figure 4), the metabolites with VIP values over 1 across the other three time intervals (T1, T2, T4) were selected as treatment−attributable metabolites. Metabolite concentrations of differential metabolites at each time interval were visualized as a heatmap (Figure 5). As expected, most selected metabolites were significantly changed between S and M group in T1, T2 and T4. At T1, treatments RPM and PPM made little change to these metabolites as compared with M group, but greatly changed from T2 onwards, reflecting the latency of drug efficacy. Treatment greatly reversed these S/M differential metabolites towards the sham group, showcasing their efficacies; in addition, treatment also greatly changed the metabolites that had no significant difference between S and M group, indicating drug bias.

### 2.6. Significant Pathways Shared by RPM and PPM Treatment Enriched by MSEA

MetaboAnalyst 5.0 “https://www.metaboanalyst.ca/ (accessed on 9 July 2022)” was used to analyze treatment−attributable metabolites selected in the previous stage to further understand the potential metabolic pathways underlying the treatment of VaD by drugs (Figure 6). The effect value from the pathway topological analysis is represented on the *x*−axis in Figure 6A, and the enriched pathway’s−log(p) significance level is shown on the *y*−axis. Each dot’s color and size were determined by the pathway effect value and the −log (*p*) value, respectively. Pathways with an impact over 0.10 was deemed as having a significant impact, shown in Table 3. The enrichment overview (Figure 6B) shows the involvement of amino acids (such as glycine, serine, threonine, tryptophan, tyrosine, alanine, aspartate, glutamine acid) metabolism, glutathione metabolism, glyoxylate and dicarboxylic acid metabolism, glycerophospholipid metabolism, TCA cycle, and the pentose phosphate pathway in the treatment of VaD, with Vitamin B6 metabolism being the most significant. Most of these pathways were closely related to VaD.

### 2.7. Hallmarks Implicated in the Efficacy of RPM and PPM on the Treatment of VaD

The bioinformatics analysis was based on a self−constructed metabolite and protein interaction network, where R was used to track down the proteins interacting with the characteristic metabolites and for a GO enrichment analysis of the genes corresponding to these proteins. The hallmark analysis of genes interacting with differential metabolites showed biological pathways such as glycolysis, oxidative phosphorylation, and inflammatory response were significantly correlated with chronic cerebral hypoperfusion injury (Figure 7A). Genes belonging to these pathways (Figure 7B) and differential metabolites were used to construct gene−metabolite interaction networks (Figure 7C) focusing on glycolysis and inflammatory response. 

### 2.8. Specifically Changed Metabolites and Associated Pathways by RPM and PPM

#### 2.8.1. Determination of RPM and PPM Specific Metabolites 

The RPM and PPM groups were compared with the S and M groups using PLSDA analysis (Figure 8A,B). There is a clear separation between the S group, M group and treatment groups. The intersection of metabolites with VIP values over 1 at T1, T2, T4 for component differing treatments from sham/model groups were treatment−specific metabolites (Figure 8C,D).

#### 2.8.2. Significant Pathways Specific to RPM and PPM Treatment Enriched by MSEA

MSEA enrichedsignificant pathways specific to RPM and PPM treatment (Figure 9). By comparison with the shared pathways (Figure 6), the specific pathways for RPM (Figure 9A) and PPM (Figure 9B) were cysteine and methionine, and riboflavin metabolism, respectively.

#### 2.8.3. RPM and PPM Specific Mechanism in the Treatment of VaD

In order to understand the mechanism of RPM and PPM in the treatment of VaD rats, RPM and PPM treatment−specific metabolites (Figure 8C,D) were mapped to the metabolite−protein network and corresponding genes were then subjected to gene enrichment analysis. To give a picture of relations among the enriched metabolic and signaling pathways, a term network was constructed (Figure 10A,B) by connecting two closely−related terms, defined as having 70% genes in common. The construction includes two steps: starting from metabolic modules (colored in red) to find related terms (in green); and further from these related terms to other related (in green). The self−built metabolite and protein interaction network was used to generate gene−metabolite−pathway networks (Figure 10C,D). 

## 3. Discussion

The prevalence of VaD is second only to that of Alzheimer’s disease, and it exhibits a wide range of vascular dysfunctional or damage−related cognitive impairment symptoms. In this study,a model of rat vascular cognitive impairment was established by 2−VO, which can better simulate the hypoperfusion and hypoxia of human brain caused by chronic cerebral hypoperfusion. The main pathophysiological basis of 2−VO is reduced glucose metabolism in brain tissue, impaired energy metabolism, neuronal defects, neurotransmitter alterations, cholinergic receptors deficiency, abnormal protein synthesis, damage to brain white matter and so on [23], which resembles the pathogenic process occurring in the brain tissue as a result of prolonged cerebral hypoperfusion. PM has been reported to have a certain therapeutic effect on VaD. Considering the great efficacy discrepancy between raw and processed PM, investigating the difference between RPM and PPM in the treatment of VaD is important, and can better guide the use of PM.

MWM has been widely used in studies on learning memory, and behavioral biology. The rich experimental parameters obtained from this experiment were analyzed to evaluate spatial learning and memory abilities of rats [24]. In the place navigation test, the escape distance and latency of rats in M group were significantly longer than those in S group in the early stage, indicating impaired spatial learning abilities of VaD rats, which was improved after drug administration (Figure 1B,C). Figure 1E,F showed that rats in M group spent less time in the target quadrant and traversed the plateau significantly less often than S group, indicating severely impaired cognitive function by2−VO. Compared with 2−VO group, rats in the drug administration group spent longer time in the target quadrant and number of times crossing the platform location were significantly increased. The results demonstrated that RPM and PPM could improve the spatial memory ability of VaD rats and greatly ameliorate their impaired cognitive performance.

The hippocampus is the most vulnerable to cerebral ischemia, which causes neuronal apoptosis and damage, and hippocampal atrophy, and as a result, cognitive impairment. The results of HE staining in the experiment (Figure 2) showed that the hippocampal structure was damaged and cell death occurred in the model group, which could be greatly improved by PPM and RPM. Water maze performance is sensitive to hippocampal damage [25]; hence, both PPM and RPM greatly enhanced space and memory abilities of VaD rats.

Systemic oxidative stress is a common feature of neurological diseases such as VaD [26], which can lead to neuronal damage and apoptosis, resulting in neuropathological damage and brain injury, and ultimately cognitive impairment [27]. Antioxidant defense systems such as GSH, SOD protect cells from ROS−related damage [28]. MDA is a significant product of the oxidative stress process and a hallmark of lipid peroxidation [29]. As compared with S group (Figure 3), the SOD activity and GSH level were significantly decreased and the MDA level was markedly increased in the M group, indicating severe oxidative stress in VaD rats and great consumption of GSH and inhibition of SOD activity. Both treatments show no obvious improvement on SOD activity but show significant decreases of MDA and an increase in GSH. RPM outperformed PPM in decreasing MDA level while PPM outperformed RPM in increasing GSH level, showcasing the discrepancy between RPM and PPM.

Numerous circulating metabolites have been linked to cognitive deterioration or dementia development and may represent promising biomarkers of preclinical dementia. This makes metabolomics a viable new strategy for determining dementia’s pathology and finding biomarkers for dementia risk [30].

In this study, metabolites associated with the treatment of VaD were significantly enriched in vitamin B6 metabolism, glutathione metabolism, amino acid metabolism, TCA cycle and pentose phosphate pathways (PPP) (Figure 6). The interconnection between the pathways can be seen in Figure 11. The most enriched pathway was vitamin B6 metabolism. Pyridoxamine, pyridoxal, pyridoxine, and their 5′−phosphate forms are all classified under the definition of vitamin B6, since they can interconvert between each other [31]. Of these, pyridoxal 5′−phosphate (PLP) is the only B6 vitamer that acts as a cofactor for enzymes [32]. The plasma PLP level of rats in group M was decreased (Figure 5), and the pyridoxal level was decreased in the early stage of the experiment and increased in the later stage, which indicated that 2−VO could cause the disorder of vitamin B6 metabolism in rats. The pyridoxal levels of each administration group were reversed towards the S group in the later period of the experiment, which suggested that RPM and PPM can all ameliorate the vitamin B6 metabolic disorder caused by 2−VO. PLP activity is required for a variety of processes, including the production of proteins and polyamines, the metabolism of amino acids and neurotransmitters, the mitochondrial processand erythropoiesis [33]. Therefore, the disorder of vitamin B6 metabolism might affect amino acid metabolism pathways such as tryptophan and glutathione in Figure 6B. PPP is associated with the production of PLP. PPP provides ribose 5−phosphate (5RP), which produces glutamine through the GS/GOGAT cycle, and then glutamine can synthesis to generate PLP [34]. In addition, PPP is the main source of NADPH. NADPH is a reducing agent that is essential for maintaining cellular antioxidant defenses such as active reducing GSH, thus PPP is critical for cellular redox homeostasis and DNA repair [35]. As a result, GSH has been used as a marker for PPP [36]. In Figure 3 and Figure 5, it could be seen that the plasma levels of GSH in VaD rats decreased after modeling and the levels increased after administration. This indicates that multiple pathways such as PPP and glutathione metabolism are affected by 2−VO, while RPM and PPM can exert their therapeutic effects on VaD rats through these pathways.

Interactions between small molecule metabolites and proteins play a critical role in regulating protein function and controlling various cellular processes [37]. By tracing protein interactions with characteristic metabolites and enriching the genes corresponding to these proteins, biological pathways such as glycolysis and inflammatory response were found to have strong correlations with VaD. The blue gene in Figure 7C is related to glycolysis, and its associated metabolites such as β−D−glucose 6−phosphate and dihydroxyacetone phosphate (DHAP) are mainly involved in the glycolysis and PPP, both of which are involved in glucose metabolism. Neuropathology, such as ischemic brain injury and neurodegenerative illnesses, is significantly correlated with dysfunction of glucose metabolism in the entire brain or in particular cell types [38]. DHAP and β−D−glucose 6−phosphate were significantly lower after modeling and their levels were increased later in the dosing period of the treatment group (Figure 5). The regulation of these metabolites in the administration group showed that RPM and PPM could provide energy by regulating glucose metabolism, thereby protecting VaD rats. The red gene in Figure 7C is associated with the inflammatory response. 2−VO leads to a decrease in cerebral blood flow, which leads to the production of excessive ROS and inflammatory cytokines in ischemic blood vessels, resulting in the inhibition of the antioxidant system and the formation of an inflammatory environment [39]. GSH plays a critical role in antioxidant defense and in maintenance of neuronal redox homeostasis [40], and it is acutely depleted during inflammation. The plasma level of PLP is negatively correlated with systemic markers of inflammation [41], and studies have shown that it can reduce inflammation in vivo by affecting the activity of the inflammasome [42]. The administration group can hence attenuate the inflammatory response by the effects on the metabolism of vitamin B6 and GSH.

In order to further explore the difference in the treatment of RPM and PPM on VaD rats, the differential metabolites were further screened for the two treatments. Compared to the enriched shared pathways (Figure 6B), the differential pathways for RPM (Figure 8A) were cysteine and methionine, and taurine and hypotaurine metabolism; while the differential pathways for PPM were riboflavin metabolism, and taurine and hypotaurine metabolism. Both RPM and PPM could affect taurine and hypotaurine metabolism. Taurine is a key functional amino acid with multiple functions in the nervous system [43]. The relationship between taurine and NADH dehydrogenase−related pathways is significant in the gene−metabolite−pathway network (Figure 10C,D) for RPM and PPM. Taurine deficiency mediates impaired complex I activity, which affects energy metabolism, mainly through an increase in the NADH/NAD+ ratio, and regulates energy metabolism through feedback inhibition of key dehydrogenases [44]. The level of taurine in the M group was markedly decreased (Figure 8C), while plasma taurine levels were significantly elevated at T2 in both the PPM and RPM groups, which suggested that modulation of taurine level is a common therapeutic effect of RPM and PPM on VaD.

Cystine and methionine metabolism was a pathway specifically enriched by RPM (Figure 9A). Homocysteine (Hcy) is a homologue of cysteine, an intermediate product of the methionine cycle [45]. With the help of vitamin B6 (a coenzyme), Hcy combines with serine to form cysteine [46]. A large amount of clinical and epidemiological data shows that abnormal Hcy level is directly related to the occurrence of different types of diseases such as cardiovascular diseases and central nervous system diseases [47]. Therefore, RPM may affect the metabolism of Hcy by regulating the metabolism of cysteine and methionine and the metabolism of vitamin B6, thereby reflecting the therapeutic effect on VaD rats.

Riboflavin metabolism (Figure 9B) is a special metabolic pathway enriched after PPM treatment. The plasma level of riboflavin in rats of M group decreased in the later stage (Figure 8D), at the same time it increased significantly in the PPM group while the effect of RPM on riboflavin was not obvious. Riboflavin is considered to be an important component of mitochondrial energy production mediated by electron transport chain, and it is particularly important for the normal production of ATP, which leads to the stabilization of cell membranes and the maintenance of adequate energy−related cellular functions. Interaction between riboflavin and flavoprotein is associated with protection of neuronal cells from oxidative stress and apoptosis [48]. Deficiency of riboflavin causes mitochondrial dysfunction, which can lead to impaired TCA cycle and reduced energy production, resulting in neurological disorders.

## 4. Material and Methods

### 4.1. Drugs and Reagents

RPM and PPM used in the laboratory were purchased from Anhui Hongkun Pharmaceutical Co., Ltd. (Hefei, China). The major ingredients of RPM and PPM were analyzed by LC−MS (Appendix A). We purchased assay kits for malondialdehyde (MDA), superoxide dismutase (SOD), and glutathione (GSH) from Jiancheng Bioengineering Institute (Nanjing, China). Methanol, acetonitrile, and formic acid were purchased from Sigma (Kawasaki, Japan).

### 4.2. Preparation of PM Extracts

RPM and PPM were extracted twice with 50% ethanol (1:10, *w/v*) under reflux, each over 2 h. A rotatory vacuum evaporator was used to concentrate the decoction until it was completely dry. Before studies, the extracts were suspended in sterile saline and kept in a refrigerator at 4 °C. The extract components of RPM and PPM were identified by LC-MS, witch were shown in Appendix A.

### 4.3. Animals and Experimental Procedure

Adult male SpragueDawley (SD) rats (180−220 g) were housed in an environment of constant temperature and humidity (temperature of 25 ± 1 °C, relative humidity of 50 ± 10%), with a 12 h light−dark cycle. Animals were allowed free access to water and food. All experiment protocols wereapproved by the Institutional Animal Care and Use Committee at the Nanjing University of Science and Technology in accordance with relevantguidelinesand regulations (Approval ID: ACUC−NUST−20211209).

The rat VaD model was established by permanent bilateral common carotid artery (2−vessel CCA) occlusion [49]. Food and water were withdrawn for 12 h and 4 h before surgery, respectively. The neck hair was shaved, a skin incision was made along the neck’s ventral midline, and the bilateral common carotid arteries were revealed by blunt dissection under isoflurane (1.5–5%) anesthesia. The bilateral common carotid arteries were separated from the vagus nerve, and double ligated with surgical sutures. Neck muscles and skin were sutured back at the end of the surgery and the wound was disinfected with 75% ethanol. A similar surgery procedure was performed without CCA ligation in the sham group.

Surviving rats after modeling were randomly divided into a sham−operated control group (S group, N = 6), a 2−VO model group (M group, N = 6), a 2−VO model + RPM treatment group (RPM group, N = 6) and a 2−VO model + PPM treatment group (PPM group, N = 6). The surgical recovery period was 4 days after the operation. On the fifth day after surgery, the treatment was intragastrically administered every day for 4 weeks. The doses (2.0 g/kg) given to animals in treatment groups were calculated as grams of crude dry extract of RPM and PPM per kilogram body weight [7]. The sham and model groups received normal saline.

### 4.4. Morris Water Maze Test

The Morris Water Maze Test (MWM) was used to assess learning and memory abilities of rats during the last 6 days of dosing. The maze was made up of a circular pool (diameter 160 cm and height 50 cm), filled with water, divided into four quadrants with markers of different shapes placed at the center of each. In the third quadrant, a circular escape platform with a diameter of 30 cm was submerged at a depth of 1 cm (target quadrant). The water temperature was controlled at 23 ± 1 °C and black edible pigment was added to the pool to make water opaque and the platform invisible. Animal behavior was videoed by a camera and analyzed by Python script developed in our laboratory using the OpenCV package.

On the day after the place navigation test, thespatial exploration test with the platform removed was conducted. After 5 days of acclimation, all rats were accustomed to swimming in the maze prior to the water maze test and were taught to escape from the water by climbing on an escape platform located in the center of one of the four quadrants of the maze. The rat was placed in the center of the maze and given 5min to explore it after a 1min adaption period. Prior to the maze test, all rats were permitted to swim in the pool for 120 s. The rats were then placed in different quadrants of the pool and allowed to search for the hidden platform (place navigation). The rats’ escape latency (the amount of time spent finding and mounting the platform in the water maze) and escape distance (the swimming path before finding the platform in the water maze) were recorded daily. If they failed within 120 s, they were placed on the platform by hand and rested for 20 s before another trial. The space exploration experiment was run after conducting the place navigation test. After the withdrawal of the platform, the trajectory of the rat reaching the platform was recorded for 120 s and the number of entries (frequency) crossing the target quadrant was analyzed.

### 4.5. Sample Collection

Blood samples were collected in 1.5 mL centrifuge tubes (containing 1% sodium heparin) sequentially at each time point (7th, 14th, 21st, and 28th day after the start of dosing) and then immediately centrifuged at 3000 rpm at 4 °C for 10 min. The supernatant was removed to obtain plasma for LC−MS analysis, which was stored at −80 °C before use.

At the end of the experiment, all rats were fasted overnight and sacrificed. Serum samples were obtained from abdominal aorta sampling. The supernatant was obtained through centrifugation (3000 rpm, 10 min, 4 °C) and stored in an ultra−low temperature refrigerator at −80 °C for biochemical assay.

### 4.6. Hematoxylin−Eosin and Nissl Staining

The brain was promptly isolated and saline−rinsed in ice−cold water. Brain tissue was fixed in 4% paraformaldehyde solution for 48 h, dehydrated in ascending alcohol series and embedded in paraffin wax. HE and Nissl stains were used to stain portions that were about 4 µm thick. Then the pathological sections were observed under a microscope, images were taken and analysed (Servicebio Inc., Wuhan, China). Quantitative analysis was carried out using Image J [50].

### 4.7. Oxidative Stress Indexes

In accordance with the instructions of the kits (Jiancheng, China), we assessed the levels of malondialdehyde (MDA), glutathione (GSH), and superoxide dismutase (SOD) activity in serum.

### 4.8. Sample Preparation for LC−MS Analysis

For LC−MS analysis, 100 μL plasma sample was taken, and 400 µL of pre−cooled methanol was added and vortexed for 30 s before freezing at −20 °C for 1 h to allow full protein precipitation, and then centrifuged at 16,000 g for 10 min at 4 °C. The supernatant was taken out and lyophilized to dryness, then dissolved with 200 µL methanol−water solution (50:50, *v/v*), vortexed for 30 s, ultrasonicated in an ice water bath for 1 min, and centrifuged at 16,000 g for 15 min at 4 °C to produce supernatant for LC/MS analysis. Additional pooled quality control (QC) samples were prepared by taking 10 μL of each plasma sample and mixing it in an Eppendorf tube. The QC samples were continuously run every fivesamples to test the stability and repeatability of the mass spectrometry.

### 4.9. LC−MS Analysis

Ultra−high−performance liquid chromatography−mass spectrometry (UPLC−MS) analysis was performed on a Triple TOF 5600+ system equipped with an electrospray ionization (ESI) source (AB SCIEX, Framingham, MA, USA). The separation was performed by gradient elution using mobile phase A (0.1% formic acid) and mobile phase B (100% methanol). Separation was achieved on a WatersAtlantis^TM^ Premier BEH C18 AX column (100 × 2.1 mm, 1.7 μm) at a flow rate of 0.3 mL/min with a gradient elution using 0.1% formic acid in water as mobile phase A and acetonitrile as mobile phase B. The flow rate was 0.3 mL/min, with an autosampler temperature of 4 °C and an injection volume of 2 μL.

Full−scan mass spectra (mass range *m/z* 50−1250) were acquired in negative ion mode with an electrode voltage of −4500 V, a declustering potential (DP) of −80 V, collision energy of −35 ± 15 eV and an ion source temperature of 550 °C. The ion source nebulizer gas (GS1), heater gas (GS2) and curtain gas (CUR) were 55, 55 and 35 psi, respectively. Automated MS/MS product ion scans were performed for the 10 most intense ions using information−dependent acquisition (IDA) in high−sensitivity mode. An automated Calibrant Delivery System (CDS)was executed every sixth injection to ensure the calibration error below 3 ppm and the accuracy of the MS and MS/MS data.

### 4.10. Data Processing

Analyst TF 1.8.1(SCIEX) was used for data acquisition and the raw MS data were converted to mzXML format using ProteoWizard v3.0.21229, which were imported into R package XCMS for peak alignment, peak extraction, peak integration and spectral library comparison to identify and annotate metabolites. Metabolic features with coefficients of variation (CV) >15% in QC samples were discarded.

### 4.11. Statistic Analysis

Pattern recognition and multidimensional statistical analysis were performed in “R” language “http://cran.r−project.org/ (accessed on 19 June 2022)”. After variance stabilization normalization (VSN) and logarithmic transformation, PCA (principal component analysis) and PLSDA (partial least squares discriminant analysis) were used to visualize the pattern of groups and calculate VIP (variable importance) values of metabolites. Metabolite set enrichment analysis (MSEA) was completed using the online toolMetaboAnalyst 5.0 “https://www.metaboanalyst.ca/ (accessed on 9 July 2022)”. GO and network analysis were performed by clusterProfiler and igraph package in R, respectively. Networks were visualized in Cytoscape.

## 5. Conclusions

In this experiment, we used a water maze test and biochemical indicators to evaluate the therapeutic effect of raw (RPM) and processed (PPM) PM on VaD rats. At the same time, experiments were conducted to investigate the changes of metabolome at different time points during VaD, and to compare the differences of metabolites between RPM and PPM treatments. Non−targeted metabolomics analysis was performed on the plasma of rats in each group at different time points. Raw and processed PM exerted comparable efficacy in the treatment of VaD and acted similarly on several metabolic pathways but also showed some differences, with the former affecting cysteine and methionine metabolism and the latter, riboflavin metabolism. The results demonstrated the feasibility of metabolomics in deciphering the mode of action of herbal therapies.

## Figures and Tables

**Figure 1 metabolites-12-01297-f001:**
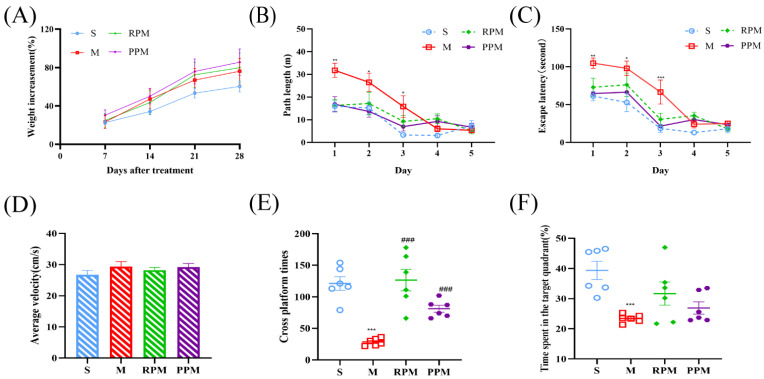
Weight increasement of rats and parameters of the MWM test. (**A**) Changes in body weight of rats after modeling and administration. (**B**) Latency to find the hidden platform in the Morris water maze. (**C**) Escape distances. (**D**) Mean swimming speed. (**E**) Target quadrant residence time. (**F**) Number of times of crossing the platform. * *p* < 0.05, ** *p* < 0.01 and *** *p* < 0.001for comparison between S and M group; ^###^
*p* < 0.001for comparison between S and M group.

**Figure 2 metabolites-12-01297-f002:**
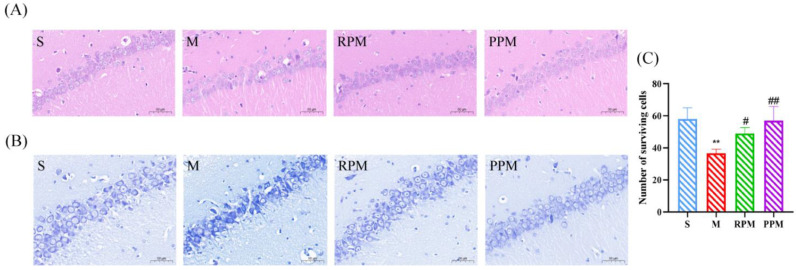
(**A**) Microscopic view of cerebral tissue stained with hematoxylin and eosin (HE). (**B**) Microscopic view of cerebral tissue stained with Nissl. (**C**) Number of surviving cells in the CA1 region. ** *p* < 0.01, compared with the control group; ^#^
*p* < 0.05, ^##^
*p* < 0.01, compared with the model group.

**Figure 3 metabolites-12-01297-f003:**
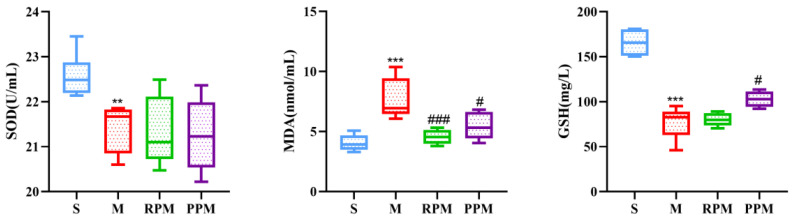
Biochemical parameters of oxidative stress. M group vs. S group: ** *p* < 0.01, *** *p* < 0.001; treatment groups vs. M group: ^#^
*p* < 0.05, ^###^
*p* < 0.001.

**Figure 4 metabolites-12-01297-f004:**
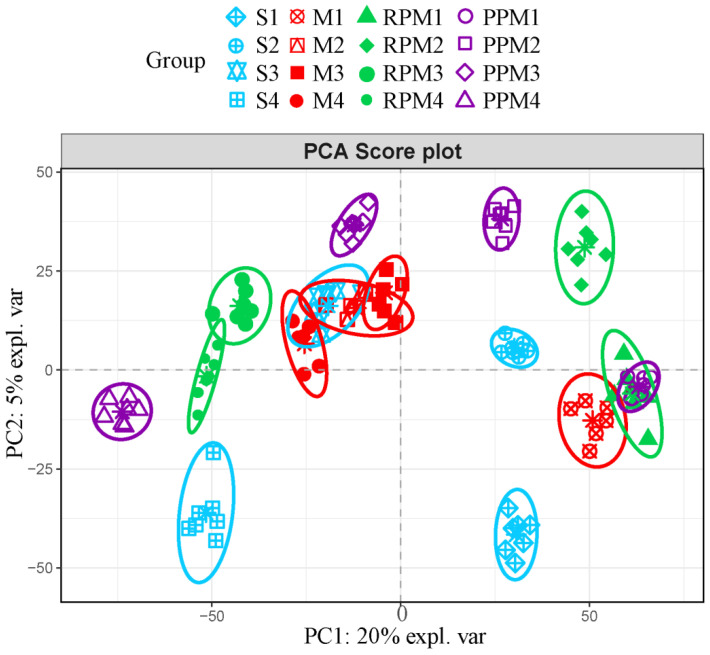
PCA scores plot for each group at different time points.

**Figure 5 metabolites-12-01297-f005:**
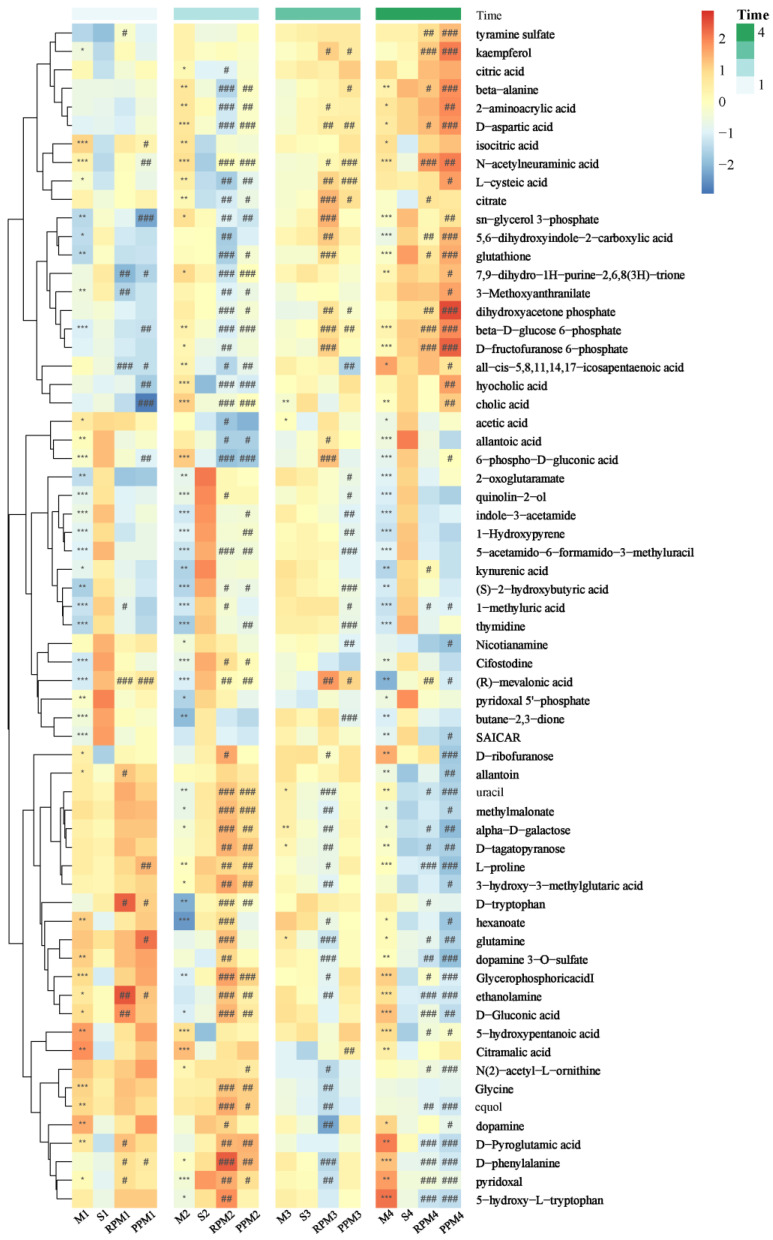
Heat map of metabolites differentiating treatment (R, P)/sham group (S) and model group (M) with hierarchical clustering on scaled median abundances of metabolites. The colors represent relative levels of metabolites as indicated in the color key, with blue for low level to red for high level. Asterisks in heat map cells indicate significant differences of metabolites between groups at each time point: marked as * for M vs. S and # for treatment vs. M. The number of marks refer to the level of significance according to *p* values with one, two or three symbols corresponding to *p* < 0.05, *p* < 0.01, and *p* < 0.001.

**Figure 6 metabolites-12-01297-f006:**
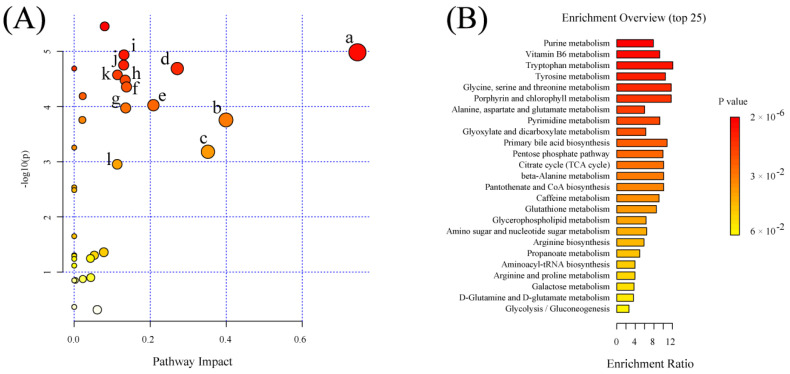
Summary of metabolic pathways significantly enriched in VaD treatment. (**A**) Bubble plot of metabolic pathways identified by MESA analysis of VaD treatment−attributable metabolites identified metabolic pathways. (a) Vitamin B6 metabolism; (b) beta−Alanine metabolism; (c) glutathione metabolism; (d) glycine, serine and threonine metabolism; (e) citrate cycle (TCA cycle); (f) pyrimidine metabolism; (g) tryptophan metabolism; (h) alanine, aspartate and glutamate metabolism; (i) glycerophospholipid metabolism. (**B**) Enrichment overview of the top 25 metabolic pathways associated with VaD treatment−attributable metabolites.

**Figure 7 metabolites-12-01297-f007:**
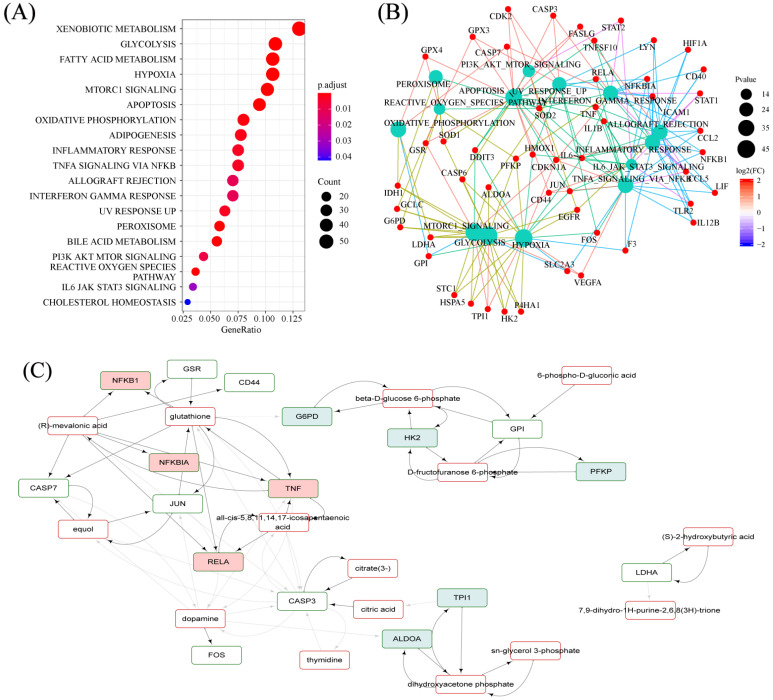
Hallmarks implicated in the efficacy ofRPM and PPM on the treatment of VaD. (**A**) Hallmarks of genes interacting with differential metabolites. (**B**) Overview of gene−pathway associations where the node size of pathway was scaled based on the *p*−value of its enrichment. (**C**) Gene−metabolite interaction networks focusing on glycolysis and inflammatory response. Gene nodes in light blue and light red belong to glycolysis and inflammatory response, respectively.

**Figure 8 metabolites-12-01297-f008:**
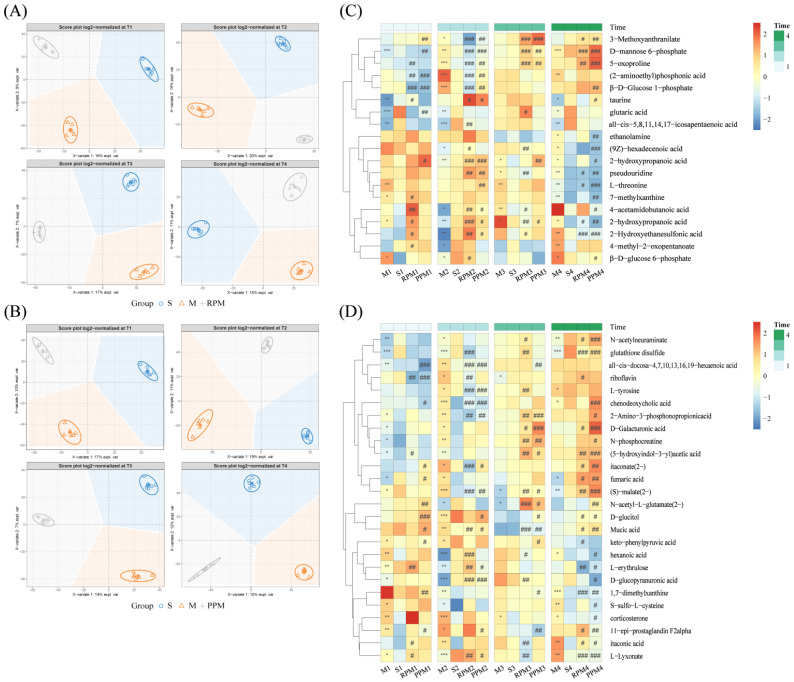
Metabolites specific to the treatment of RPM and PPM. PLSDA scores plotted for groups of sham, model and treatments with (**A**) RPM and (**B**) PPM at each time point. Heatmaps of (**C**) RPM and (**D**) PPM specific metabolites. M group vs. S group: * *p* < 0.05, ** *p* < 0.01, *** *p* < 0.001; treatment groups vs. M group: ^#^
*p* < 0.05, ^##^
*p* < 0.01, ^###^
*p* < 0.001.

**Figure 9 metabolites-12-01297-f009:**
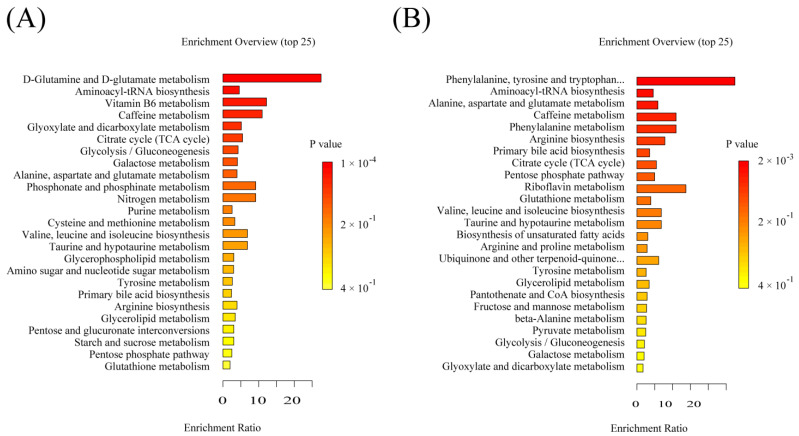
Enrichment overview of the top 25 metabolic pathways associated with (**A**) RPM and (**B**) PPM specific metabolites.

**Figure 10 metabolites-12-01297-f010:**
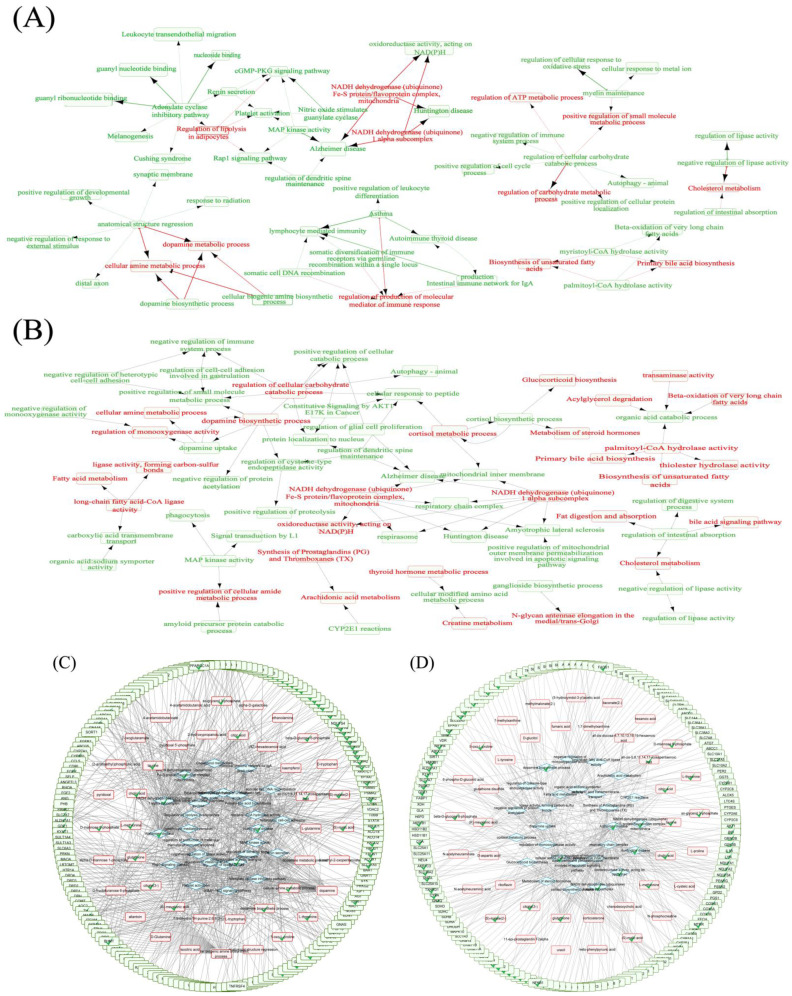
Metabolic and signaling pathways related to differential metabolites of (**A**) RPM and (**B**) PPM in the treatment of VaD. The terms in red were metabolic modules, starting from these, the arrows point to the closely−related terms in green (70% of genes in common). Gene−metabolite−pathway network of (**C**) RPM and (**D**) PPM where differential metabolites, metabolite interacting genes, gene associated metabolic or signaling pathways were framed by red, green and blue colors, respectively. The tagged nodes with the check mark are present in both RPM and PPM.

**Figure 11 metabolites-12-01297-f011:**
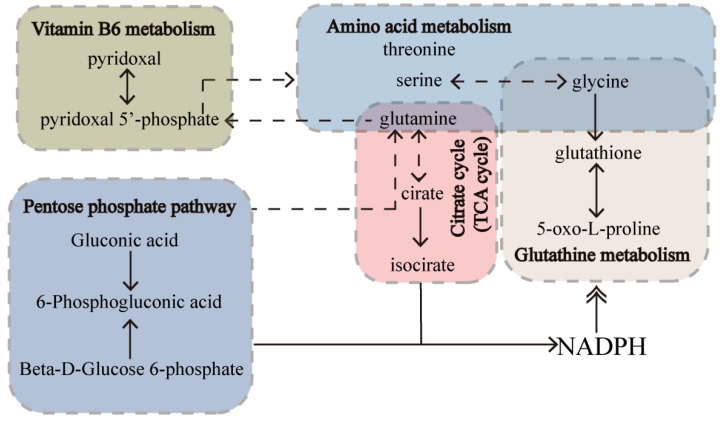
Altered metabolites and pathways implicated in VaD that could be ameliorated by both treatment of PM in rats plasma.Relationships that are direct or indirect are denoted by solid or dotted lines, respectively.

**Table 1 metabolites-12-01297-t001:** Path length and time spent in the target quadrant (mean ± SEM).

Group	Path Length (m)					Time Spent in the Target Quadrant (%)
	Day 1	Day 2	Day 3	Day 4	Day 5	
S	15.99 ± 2.26	15.13 ± 2.66	3.32 ± 0.87	3.05 ± 0.69	7.33 ± 2.32	39.37 ± 3.01
M	31.77 ± 3.13 **	26.49 ± 4.01 *	15.82 ± 4.79 *	6.02 ± 1.28	5.37 ± 1.15	23.43 ± 0.56 ***
RPM	16.32 ± 2.55	17.15 ± 4.89	9.33 ± 2.47	10.45 ± 2.09	5.15 ± 1.38	31.70 ± 3.85
PPM	16.81 ± 3.38	13.64 ± 2.51	7.00 ± 1.99	9.26 ± 2.47	6.75 ± 1.68	26.92 ± 2.03

* *p* < 0.05, ** *p* < 0.01, *** *p* < 0.001 for comparison between S and M group.

**Table 2 metabolites-12-01297-t002:** Escape latency, average velocity and cross platform times (mean ± SEM).

Group	Escape Latency (Second)					Average Velocity (cm/s)	Cross Platform Times
	Day 1	Day 2	Day 3	Day 4	Day 5		
S	61.31 ± 6.16	52.82 ± 12.03	18.62 ± 5.19	13.00 ± 2.60	18.06 ± 5.39	26.72 ± 1.44	121.2 ± 10.76
M	104.8 ± 6.78 **	98.00 ± 9.54 *	66.55 ± 15.83 ***	24.05 ± 0.69	25.05 ± 3.16	29.35 ± 1.66	28.33 ± 2.29 ***
RPM	72.99 ± 11.99	74.21 ± 15.43	30.53 ± 8.08	35.47 ± 4.53	17.83 ± 3.28	28.26 ± 0.86	126.5 ± 17.09 ^###^
PPM	70.41 ± 12.21	66.41 ± 11.02	21.57 ± 4.33	30.19 ± 7.53	23.49 ± 5.30	29.23 ± 1.14	81.17 ± 5.54 ^###^

* *p* < 0.05, ** *p* < 0.01, *** *p* < 0.001 for comparison between S and M group. ^###^
*p* < 0.001 for comparison between treatment and M group.

**Table 3 metabolites-12-01297-t003:** Significant pathways with major change revealed by MESA analysis based on the pathway impact and *p* value.

Pathway Name	Matched Metabolites	Raw p (×10^−3^)	−log10(p)	FDR (×10^−3^)	Impact
Vitamin B6 metabolism	2/9	0.010	4.984	0.116	0.745
beta−Alanine metabolism	2/21	0.175	3.757	0.425	0.399
Glutathione metabolism	3/28	0.660	3.180	1.403	0.351
Glycine, serine and threonine metabolism	1/34	0.020	4.689	0.116	0.271
Pentose phosphate pathway	3/21	0.095	4.024	0.292	0.208
Glyoxylate and dicarboxylate metabolism	5/32	0.044	4.356	0.166	0.138
Citrate cycle (TCA cycle)	2/20	0.106	3.975	0.300	0.135
Pyrimidine metabolism	4/39	0.033	4.479	0.141	0.134
Tryptophan metabolism	1/41	0.012	4.937	0.116	0.131
Tyrosine metabolism	2/42	0.018	4.752	0.116	0.130
Alanine, aspartate and glutamate metabolism	4/28	0.027	4.575	0.129	0.114
Glycerophospholipid metabolism	3/36	1.114	2.953	2.229	0.113

## Data Availability

The data presented in this study are available on request from the corresponding author due to privacy or ethical restrictions.

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
