# Peer review of "Comparative Investigation of Raw and Processed Radix Polygoni Multiflori on the Treatment of Vascular Dementia by Liquid Chromatograph−Mass Spectrometry Based Metabolomic Approach"

_metabolites, 2022, doi:10.3390/metabo12121297_

Round 1
Reviewer 1 Report
Review of the study: ” Comparative Investigation of Raw and Processed Radix 2 Polygoni Multiflori on the Treatment of Vascular Dementia by 3 Liquid Chromatograph-Mass Spectrometry Based Metabolomic 4 Approach „
1. Introduction No information on the methods of treatment used so far.There is no justification why Radix Polygoni Multiflori should be used in this type of dementia
No clear work aims. A description of the study is not needed in the introduction. This should be moved to the methodology section
2.results:
1. methodology should be written in the methodology. This section should include the results discussed
3.1. Behavioral test - is better to put these results in a table to make them more readable
3.3. Biochemical assays - if we write that the levels were significant, how much was p?
1. Significant pathways with major change based on the pathway impact and P value – the title should specify the phenomenon under investigation
3.7. Metabolite-protein network construction - . Then, we 313 used Cytoscape to analyse the network, finding 50 genes with strong potential correlation 314 on the basis of degree of node – showed a significant correlation (p=?)
Reviewer 2 Report
In presented paper, the authors deal with interesting problems of standardization of use Radix Polygoni Multiflori on the treatment of vascular dementia. For that preps they used rats as model systems and analyzed plasma by HPLC MS/MS to obtain metabolic profiles of treated animals. They gave a nice introduction to the topic and the applied methodology. The main focus of my work is to consider changes in metabolic pathways based on the results of multivariate analysis and effect of Polygonum Multiflorum extracts on rats with induced vascular dementia.
The authors the writing style was quite challenging for me. During the preparation of the samples for the recording of the serum samples, no internal standard was used, which would be used to normalize the obtained surfaces which is typically for this technique. For this reason, I would like additional validation of statistical data, e.g. permutation or some other test, in order to further support their results.
Nevertheless, I recommend this paper for publishing after minor revision.
Reviewer 3 Report
This is an interesting article providing important insights into the therapeutic effect of raw and processed Radix Polygoni Multiflori nootropic in a rat model of vascular dementia, such as the permanent bilateral common carotid artery occlusion. By using behavioral, biochemical, and non-targeted metabolomics analysis the Authors demonstrated that, although affecting different metabolic pathways, both raw and processed extracts exerted comparable efficacy in the treatment of vascular dementia.
Although the manuscript is interesting, there are some major concerns that need to be addressed to further strengthen this research article.
Major comments
1) Abstract: It is not clear from the abstract the exact timing of the experimental protocol. Did the rats undergo 2-vessel occlusion surgery before or after the start of the 28-day treatment? Also, the Authors reported “plasma was collected once a week” but they should better explain, at least in the “2.5. Sample collection section”, the blood sample collection method for this part of the experimental protocol.
2) Introduction section - Lines 60-69: The authors reported here the abstract adding also material and methods. Please, add more clearly the main aim of the study.
3) A flow chart of the experimental protocol could enable the readers to better understand the study design. Please add.
4) Materials and methods section - 2.3. Animals and experimental procedure: In addition to the flow chart, the authors should report here more detailed information regarding the start of the treatment.
5) Materials and methods section - 2.11. Statistic analysis: Were all data normally distributed? If not, which test was used when the normality test failed? Also, which statistical test was used for behavioral or biochemical analyses?
6) Results section: Please substitute the title of each paragraph of the results with a concise and precise sentence about the main finding.
7) Results section - Line 211: Although the Authors reported “In contrast, rats in R, and P groups spent more time in the target quadrant”, this sentence seems to be not justified from the results. Please discuss.
8) Results section - Line 224: Although the Authors reported “The CA1 region is very sensitive to ischemic injury and is closely related to cognitive function”, they show histological staining on the CA3 region. Please discuss. Also, have the Authors performed Nissl staining to detect surviving neurons?
Minor comments
- Lines 14; 43; 48; 64: Please correct typos and punctuation mistakes. Please revise the entire manuscript.
- Lines 45-46: Please insert bibliography.
Round 2
Reviewer 1 Report
Thank you very much for the corrections made and I have no further commentsReviewer 3 Report
The Authors did a great job to address all the concerns raised by this Reviewer.